# Open Tools for Analysis of Elements Related to Public Transport Performance. Case Study: Tram Network in Bucharest

**Liliana Andrei** * **and Oana Luca**

Faculty of Civil, Industrial and Agricultural Engineering, Technical University of Civil Engineering, 020396 Bucharest, Romania; oana.luca@utcb.ro
* Correspondence: liliana.capra@phd.utcb.ro

**Abstract:** The present paper aims to be useful for public transport operators and municipalities, as it should enable them to make decisions and to optimize public transport schedules during peak hours. In this study, we outline the data and the means necessary for the creation and use of a specific database for a dynamic spatial analysis of the public transportation network. This will facilitate the analysis of public transport vehicle operating programs and the simulation of new transport programs using open-source software. This paper delivers the first digital map of the public transport in Bucharest. Using the QGIS software and the PostgresSQL database, (i) we analyzed the accessibility of public transport stops for residential areas (5-min isochrones, corresponding to walking distances of 400 m), and (ii) we determined the correlation of transport vehicle programs with the existing transport network to optimize the headway of vehicles. These two elements were considered for the analysis of public transport performance. The research study was based on the tram network in Bucharest, but it can be easily upscaled for the entire public transport network and may be replicated in other large cities.

**Keywords:** public transport; GIS; digital map; simulation; optimization

## 1. Introduction

### 1.1. Motivation

The sustainability of cities depends, to a large extent, on urban transport, which has many negative impacts such as traffic congestion, air pollution, health issues, and decreased traffic safety. In this context, public transport is a viable alternative to private car transport even for medium and long distances. Urban areas with a good public transport supply also favor pedestrian and bicycle travel [1] when safe and comfortable infrastructure is provided [2]. In many countries, the authorities have made efforts to solve urban transport problems by building more infrastructure; introducing more traffic signalization; introducing traffic management, especially for car flows; enhancing the quantity and quality of information (ITS); introducing road pricing; introducing subsidies to make public transport more attractive; and better integrating urban transport with land-use planning [3].

Measuring the performance of public transport is essential for transport authorities, enabling them to verify whether services are provided efficiently and effectively, to identify areas where performance may need to be improved, to ensure user satisfaction, and to support decision makers in making informed decisions [4].

For tracking all these factors in a fast and coherent manner, the rapid digitization of the public transport network is necessary. Digital maps allow transport authorities and/or operators to analyze, forecast, and simulate the evolution of urban public transport in correlation with spatial development, in the context of developing new forms of mobility, such as micro-mobility and the inevitable introduction of autonomous and connected urban public transport.

The digitization of the transport network by integrating the road network, stations and terminals, lines, transport schedules, the number of passengers, and accessibility is essential for carrying out analyses leading to the better adaptation to demands, increased performance and, thus, increased attractiveness of the urban public transport system. In recent years, special attention has been paid to the use of geographical information systems (GIS) for modeling mobility and transport studies [5–7].

### 1.2. Brief Literature Review

In [8], the use of GIS in Utrecht for developing a potential map of future public transportation and supply centers was discussed. The hypothesis of the study was that models in the fields of geography and planning that were integrated into GIS would produce a general picture of accessibility. A new facet of research is outlined in [9] showing the potential user impacts of introducing smartphone-generated and analog-delivered schematic bus maps in Dhaka, Bangladesh.

Numerous tools for the accessibility analysis of public transport networks have been developed: a raster-based model of the accessibility of workplaces based on public transportation services [10]; a GIS-based tool developed to allow the rapid analysis of accessibility based on different transport modes using generalized cost to measure transport costs across networks, including monetary and distance components; a tool that was demonstrated in London, UK [11,12]; a public transport criterion matrix based on existing performance standards, applied to study the public transport networks in three Australian cities, Stonnington, Bayswater, and Cockburn, representing a series of land use and transport policy backgrounds [13]; and the accessibility Interactive Visualisation Tool used in Rome to generate maps of the level of perceived accessibility of different transportation and urban facilities [14]. Consistent multilevel research has been performed for studying the hierarchical process for optimizing bus stop locations in the context of fast-growing multi-modal transit services, with a case study in Wuhan, China [15]; the use of geographic information systems (GIS), particle swarm optimization (PSO), and a genetic algorithm (GA) to model the locations of bus stops in Amman city in Jordan to determine the optimal travel times and serviceability of stops [16]; alternative optimization for the locations of bus stops presented through an optimum proposed model, applied in Sakarya (Turkey) [17]; a bi-level optimization model for locating bus stops to minimize the social cost of the overall transport system [18] and for bus stop time models; and total bus stop time models for bus stops located at mid-blocks and near intersections, based on multivariate regression analysis using an ordinary least squares method that could potentially be used to improve scheduling and transit bus system planning in a dense urban area [19]. Moreover, studies have outlined the analysis of bus dwell times using archived automatic vehicle location (AVL)/automatic passenger counter (APC) data reported at the level of individual bus stops [20], statistical distributions describing and explaining dwell time variability [21], and a method with which to identify bus stops that may need redesigning to reduce the time lost in arriving and departing [22].

The optimization of public transport routes was addressed in [23] for developing a framework for bus network optimization based on geographical information systems (GIS) and genetic algorithms (GAs), and in [24] for describing a methodology for modelling and scaling down a street network to facilitate the optimization of public transport networks, applied to the bus network of Greater Nottingham in the UK. Developing a procedure for generating input datasets for urban transit routing problem (UTRP) algorithms together with developing an interface between the UTRP and the professional transport modeling software PTV Visum for a multi-modal public transport network was performed in [25], the adaptation of a node-based optimization procedure to work with zone-to-zone trips based on a hybrid approach to calculate zone-to-zone journey times through the use of node-based concepts was included in [26], and network analysis tools for public transport optimization with a case study in Lódź, Poland, were described in [27]. Based on the three-

dimensional magic cube method [28], transport planning has been studied for determining final land suitability.

The spatio-temporal data related to transport have experienced rapid growth in recent years, and the management and analysis of the data have become dominant. The use of relational databases for this purpose has been researched for more than two decades, with many authors [29,30] proposing several models for designing a spatially enabled database management system. More recently, a model [31] has been proposed for the calculation of public transport accessibility with a relational database, using a PostrgreSQL relational database for a data processing system for large car services [32].

The movement of data has been approached in several scientific papers, with the focus on the spatial and temporal properties of individual trajectories [33], while the consideration of vehicle speed and trajectory was researched in [34,35].

The literature emphasizes that GIS shows huge potential for visualizing, managing, and modeling geo-spatial data [36,37] and can be used to evaluate the transport system, make decisions, and create and analyze different scenarios, depending on the variation of different parameters. The use of open-source software such as OpenStreetMap, QGIS, PostgreSQL, and PostGIS (for connecting relational databases to GIS) allows the integration of geospatial databases with non-spatial data necessary for modeling the transport system.

The GIS-based accessibility model developed by Liu and Zhu [38] provides a general framework for the integrated use of GIS, travel impedance measurement tools, and accessibility measures to support the accessibility analysis process. A GIS-based land use and public transport origin-based accessibility indexing model [39] has been developed for measuring accessibility based on both actual walking distances and public transport travel time. In addition, a GIS-based approach for assessing pedestrian accessibility in urban areas, with a focus on the accessibility of public transport stops and stations, was developed as an integrated approach to urban planning and mobility planning [40]. The possibility of using general transit feed specification (GTFS) to identify deviations of public transport in morning hour traffic was examined in [41], and a method for evaluating the quality of pedestrian paths and the accessibility of railway stations was developed in [42]. A real-time tactic-based control (TBC) procedure for increasing the service reliability and actual occurrence of synchronized transfers in a headway-based public transport system was detailed [43], using selected online operational tactics, such as holding, boarding limits, and skipping stops, all of which are based on real-time data, for minimizing the additional travel time for passengers and reducing the uncertainty of meetings between public transport vehicles.

GIS was also used to assess the effectiveness and sustainability of public transportation. A simple GIS-based tool that allows the rapid analysis of accessibility via different transport modes in London, using ArcGIS, is presented in [11]. For measuring the sustainability of urban transport performance, [44] proposes a performance model with which to evaluate the development of urban transport in terms of sustainability. This model is based on indicators such as traffic congestion, traffic pollution, and land consumption for transport infrastructure.

It may be concluded that using GIS for digital maps, the optimization of bus location and public transport routes, GIS-based accessibility models, effectiveness, and the sustainability of public transport, together with research on data and data movement, are subjects extensively discussed in the scientific literature. However, the use of open tools in providing digital maps for public transport, and for the analysis of transport performance through the accessibility of public transport stations for residential areas and the optimization of the headway of vehicles is still a work in progress and may be completed with new elements.

### 1.3. Objectives of the Paper

Taking into consideration the above conclusion, we propose the following research question: how can we support decision making for increasing the performance of public transport in an attempt to increase the attractiveness of public transport?

We selected Bucharest as the case study for answering the research question because recent studies [45] have shown that, in this city, the largest urban agglomeration in Romania, emissions from road traffic (PM10 and PM2.5) have become a major threat to air quality. In addition, traffic congestion has led to negative impacts on the economy, increased travel times and fuel consumption, a decrease in the reliability of cars due to frequent acceleration and braking, and a lack of accessibility for emergency vehicles (ambulances, fire engines, and police vehicles) [46]. Under these circumstances, more sustainable transport measures, such as public transport, must be considered by citizens.

Bucharest has 770 km of bus network, organized into 104 lines; 282 km of tram network, organized into 24 lines; and 142 km of trolleybus network, organized into 17 lines, with a density of about 3.99 km/km$^2$. For each public transport mode, there are 2288 stops/stations, as follows: 605 tram stations, 1385 bus stops, 57 trolleybus stops, and 241 common bus and trolleybus stops. The average distance between tram stations is 0.466 m; it is 0.497 m for bus stops and 0.483 for trolleybus stops. The average daily operating fleet is 1501 vehicles (266 trams, 1071 buses, and 165 trolleybuses). The number of passengers per day is 1.95 million, with a modal split of 47.08% for trams, 47.18% for buses, and 9.74% for trolleybuses [47]. To maximize the ease of use and efficiency of the public transport system in Bucharest, the fare systems have recently been integrated [48]. However, the public transport lacks timetable integration. In addition, even though investments have been made for the partial replacement of the public transport fleet of vehicles in recent years, Bucharest's surface public transport still lacks attractiveness.

To respond to the research question, the first step was working, with open tools, on a digital map of public transport, which is not currently available for the surface public transport in Bucharest (or in other Romanian cities), despite the fast digitization process initiated by the public administration due to the COVID 19 pandemic [49].

The second step was to explore the analysis of public transport performance supported by the elements related to (i) the accessibility of public transport and (ii) optimizing the headway of vehicles. According to the service contract, the public transport operator in Bucharest evaluates public transport performance based on 11 indicators [50]: (1) the trips that are canceled or irregular due to operator fault; (2) routes canceled for a period of 24 h due to operator fault; (3) the number of passengers affected by the trips or routes canceled due to operator fault; (4) the total number of vehicles used daily, compared to the number of vehicles necessary for the realization of the schedule (weight: 5.0%); (5) complaints regarding the quality of transport; (6) environmental protection; (7) vehicles (the average age of the vehicles in the operating fleet and comfort requirements); (8) the compensation paid by transport operators for non-compliance with quality and environmental conditions regarding transport; (9) the number of violations found and sanctioned by authorized personnel regarding non-compliance with legal provisions; (10) the number of traffic accidents due to operator staff; (11) an index of passenger satisfaction. The main weight was given to trip indicator (1), which is related to schedule adherence (39%); however, no clear details are provided on the specific methodology used for assessing public transport performance.

In summary, to answer the research question for the Bucharest case study, the objectives of the study were 1. to develop a digital map of the Bucharest public transport network, supported by an open-source GIS application, and 2. to use the digital map to explore the elements contributing to public transport performance in Bucharest, Romania, namely, 2a, the accessibility of public transport stations in 400 m isochrones (5-min walking distances) for residential areas, more specifically presenting the analysis of the correlation of the transport schedules of the tram network in Bucharest, and 2b, the optimization of the headway of vehicles using the QGIS software and PostGIS database, allowing the simulation of vehicle movement in the existing urban public transportation network and

improving the public transport operation schedule, thus avoiding crowding in public transport stations.

The rest of the paper is organized as follows: First, in Section 2, the materials and methods utilized are detailed. Second, in Section 3, the results and discussion are presented, including the digital map of public transport in Bucharest, outlining the accessibility of public transport for residential areas, and optimizing the headway of vehicles using the QGIS software and the PostGIS/PostgreSQL database. Last, Section 4 presents a short conclusion based on the results and discusses future endeavors.

## 2. Materials and Methods

The methodology for supporting decision making for increasing the performance of public transport was created following the scheme in Figure 1.

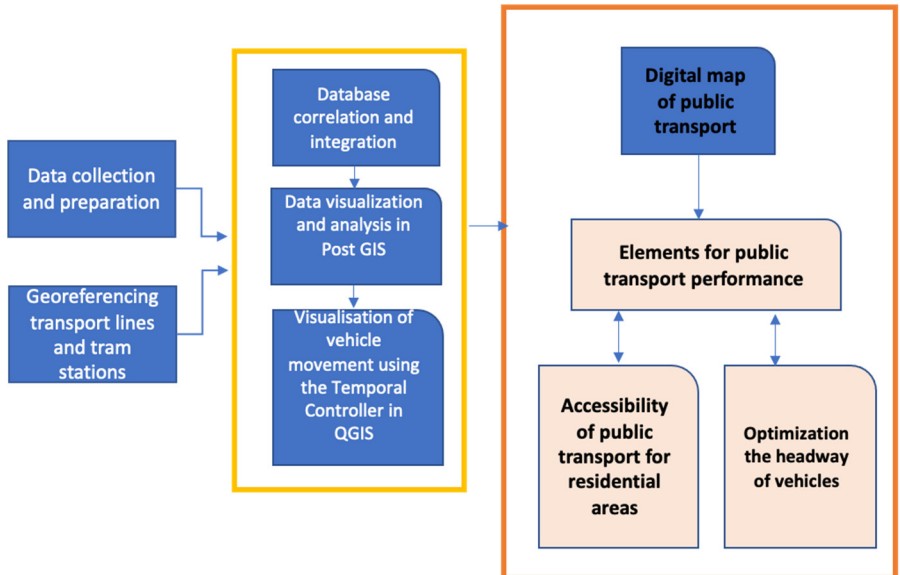

**Figure 1.** Logical scheme for the methodology development.

### 2.1. Tools, Data Collection, and Preparation

The method for the present research was carried out using open-source software (OSS), which has numerous advantages such as accessibility, flexibility, cost-efficiency, and the ability to be run on different operating platforms. The OSS used was QGIS 3.16, PostgreSQL 13, pgAdmin4 as a development and administration platform, and PostGIS Bundle 3 for PostgreSQL. The QGIS plugins used for this work were the QuickOSM plugin, for extracting the roads, railways, buildings, and administrative unit objects from OpenStreetMap (OSM) [51], and the DBManager plugin, for querying and managing the database. PgAdmin was used as an administration platform for PostgreSQL, and PostGIS provided support for the geographic objects, allowing location queries in the PostgreSQL database.

Bucharest population data were collected from the National Institute of Statistics, 2011 National Census [52], and Bucharest Regional Directorate for Statistics [53]. Data on routes, stations, transport lines, operating schedules, tour start and end times, and the headways of vehicles were retrieved from the website of the main public transport operator and the transport authority [54–56].

We used three types of data to develop the methodology: stop locations, the stop sequence for each line and stop arrival times.

### 2.2. Characteristics of the Population in the Study Area

The data necessary for developing the model relate to the population number in the city, density, and building area, together with a map of the residential areas in the city.

Bucharest city has a population of about 1.9 mil. inhabitants, a density of 8771 inhabitants/km$^2$ [53], and a building area representing about 70% of the entire city surface.

We used the residential area layer (extracted from OSM in Figure 2), indicating that these are generally uniformly distributed over the territory. We can observe that the density of the population varies between the districts (Table 1). Based on this map, we can estimate the penetration of public transport in the study area and the accessibility of public transport using the 400 m isochrone around the stops/stations.

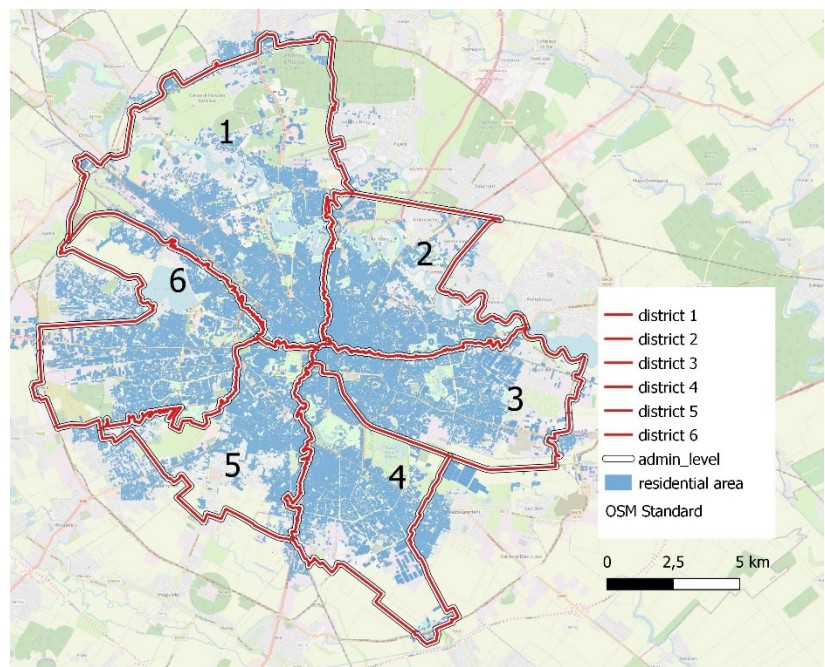

**Figure 2.** Administrative area and residential zones of Bucharest (© OpenStreetMap contributors).

**Table 1.** Population density per district [53].

| District | Density (Inhabitants/km$^2$) |
|---|---|
| 1 | 3.548 |
| 2 | 11.604 |
| 3 | 13.915 |
| 4 | 9.501 |
| 5 | 10.322 |
| 6 | 95.08 |

### 2.3. Georeferencing

To facilitate the analysis, we considered the representation standards for road networks for the composition of the database.

Currently, the most used format for public transport data is the General Transit Feed Specification (GTFS), a data standard developed by Google and used to describe a public transport system. The main purpose of the GTFS is to allow transit operators to upload programs to Google Transit so that Google Maps users can easily view transportation modes, schedules, fares, etc.

Creation of Digital Map Based on Transport Routes and Tram Stations in Bucharest

Elements of the transport network include fixed physical elements, such as the public transport network (lines, routes, infrastructure, stations, parking stations, and ticket/subscription selling points), and variable elements (vehicles, schedules, rates, transport capacities, and the number of passengers). We correlated the databases for lines, stations, and schedules to obtain an overview of the system. The transport lines were referenced in relation to the street layer and railway layer extracted from OSM and correlated with the information from the operator's website [54] about public transport lines.

For georeferencing the tram stations, we defined attribute tables (see Table 2), considering the structure of the GTFS data [57], whose format and structure were used. The localization of the stations on the map was performed manually, based on OSM and using smartphone GPS data collected directly in the field. This method was used because of the lack of availability of latitude and longitude data (Figure 3).

**Table 2.** Format and contents of the "stops" attributes table [57].

| Field Name | Type | Required | Description |
|---|---|---|---|
| stop_id | ID | Required | Identifies a stop or station. |
| stop_name | Text | Conditionally Required | Name of the location. |
| stop_lat | Latitude | Conditionally Required | Latitude of the location. |
| stop_lon | Longitude | Conditionally Required | Longitude of the location. |

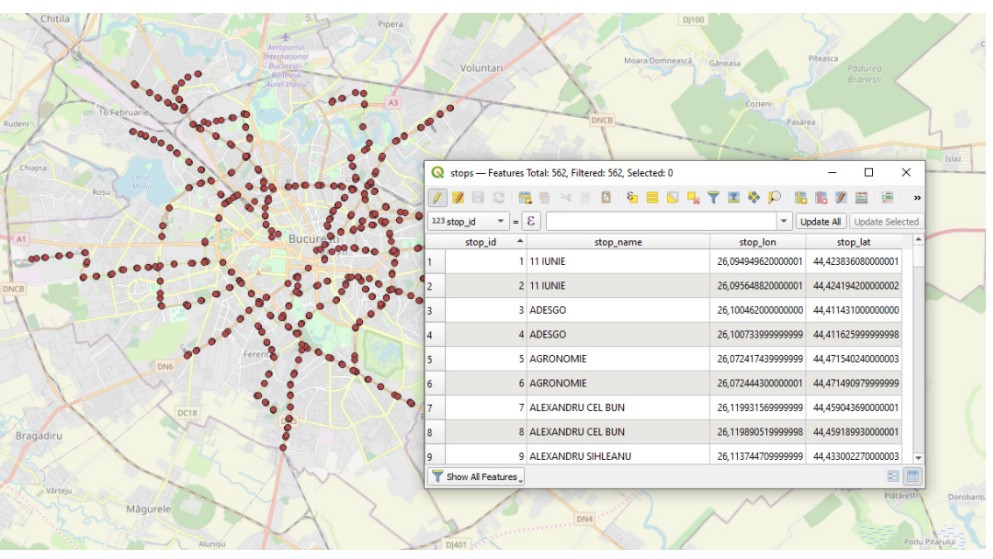

**Figure 3.** Tram station georeferencing (station positions are represented by dots on the map, and the attributes table contains their IDs, names, and geographical positions) (© OpenStreetMap contributors).

### 2.4. Database Model Development

We rebuilt the data (station positions, vehicle schedules, trips, stop sequences, stop times, and arrival times) in spreadsheets, one for each tram line, to enable them to be easily imported into PostGIS/PostgreSQL. This allowed the simulation of further changes in the vehicle schedule per line.

### 2.5. Database Correlation and Integration

Working with a large volume of data that exceeded the capabilities of a typical spreadsheet led us to conclude that a combination of PostgreSQL, PostGIS, and QGIS was necessary, as working only with the local file might have been very slow or even impossible when transforming the data into animations.

We performed the correlation and integration of the data directly from QGIS by creating a new connection in the browser PostGIS, and, once the connection had been completed, we imported both spatial and non-spatial files into this database via DBManager (QGIS).

### 2.6. Data Visualization and Analysis in PostGIS/PostgreSQL

To create the view and animation in QGIS, we combined the tables in the database into a single table. The resulting trips table contained more than 120,000 records, only for the tram network. With the addition of the bus and trolleybus network, the number of entries would exceed 2 million, the database becoming quite extensive, making it necessary to support QGIS operations using PostGIS/PostgreSQL. Adding an index to the records in the database will help in improving the performance of queries, accelerating the retrieval of data, and ensuring uniquely identifiable records in the database. For the purpose of this study, only the visualization of the data was necessary. Loading stations were checked in the preview tab in DBManager or in pgAdmin (Figure 4).

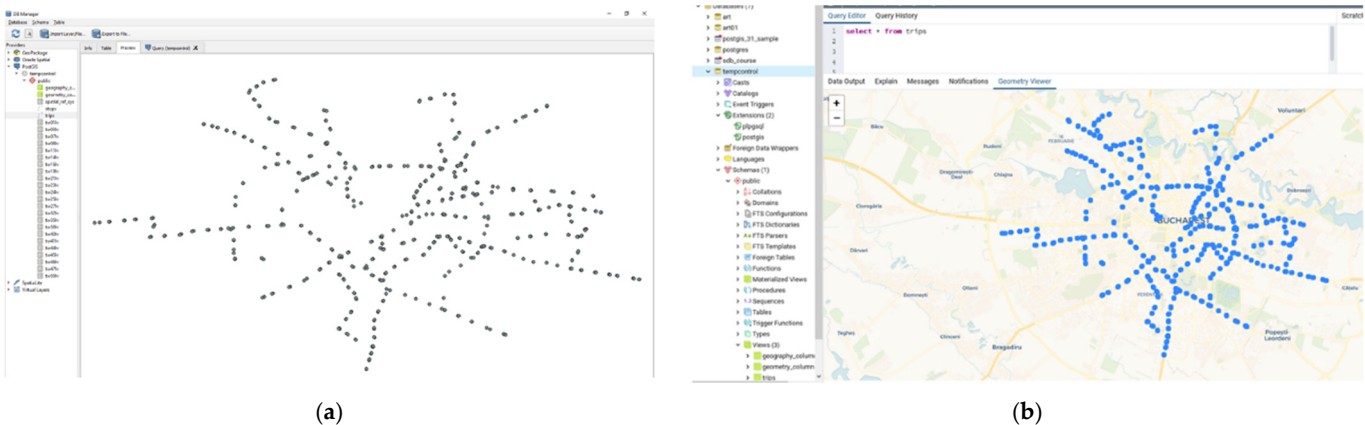

(**a**)                                                                                                                                  (**b**)

**Figure 4.** Previewing the vehicle position in stations. (**a**) In DB Manager. (**b**) In pgAdmin using the Geometry Viewer.

## 3. Results and Discussion

### 3.1. Digital Map of Public Transport

The transport lines and stations were georeferenced in relation to the street layer [51] extracted from OSM and correlated with the information found on the operator's website [54] about the lines and the approximate positions of the stations' locations on the city map.

The map in Figures 5 and 6 confirms that the Bucharest city area and, especially, the residential areas are well covered by the surface public transport.

The tram network (density: 1.18 km/km$^2$) has a radial network consisting of only one circular line (inner ring) and is remarkably extensive (Figure 7), but it is discontinuous in the central area of the city. Based on the present map, a fast decision on reconnection may be taken by the public transport operator when referring to the redesign of the tram transport schedule.

Since the accessibility of the public transport network influences travel demand and urban development [58], its evaluation is necessary for determining the performance of the public transport system. In Figure 8, we show the determination of the isochrones at a walking distance of 400 m (5-min isochrones) from each stop/station using the buffer feature of QGIS based on a fixed distance, considering an average pedestrian speed of 4.8 km/h. For the present research, we used radial isochrones, taking into consideration the fact that Bucharest has a plain topography, does not change much in elevation, has facile connections between main roads and public transport, and has dense residential areas.

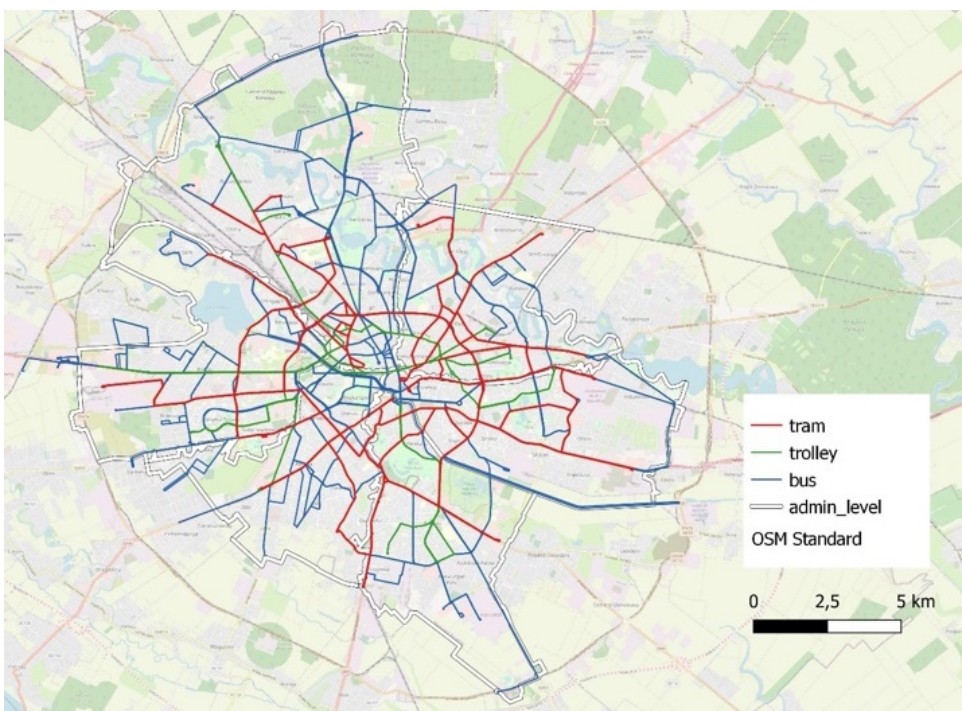

**Figure 5.** Administrative area and public transport network in Bucharest (blue line—bus network; green line—trolleybus network; red line—tram network) (© OpenStreetMap contributors).

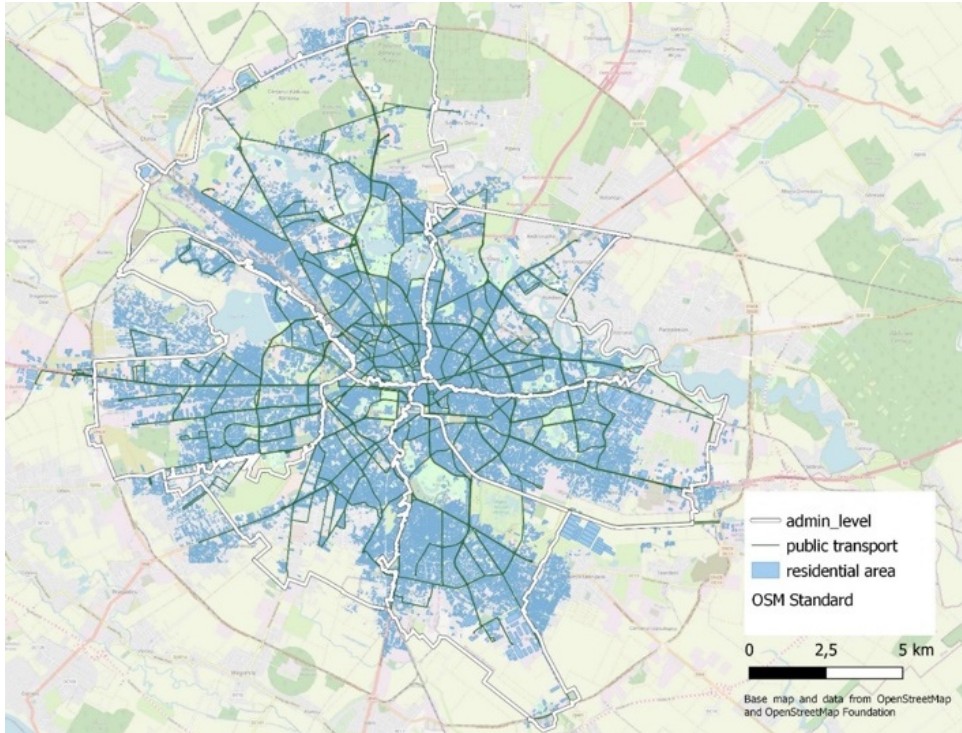

**Figure 6.** Residential areas and public transport network (© OpenStreetMap contributors).

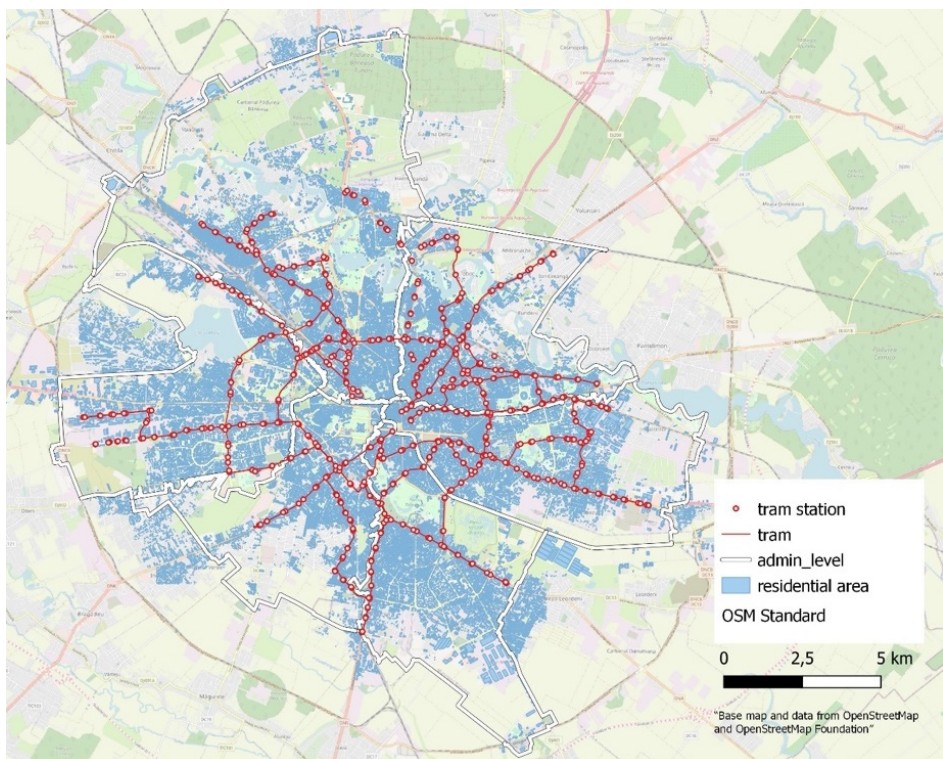

**Figure 7.** Tram network and tram stations (© OpenStreetMap contributors).

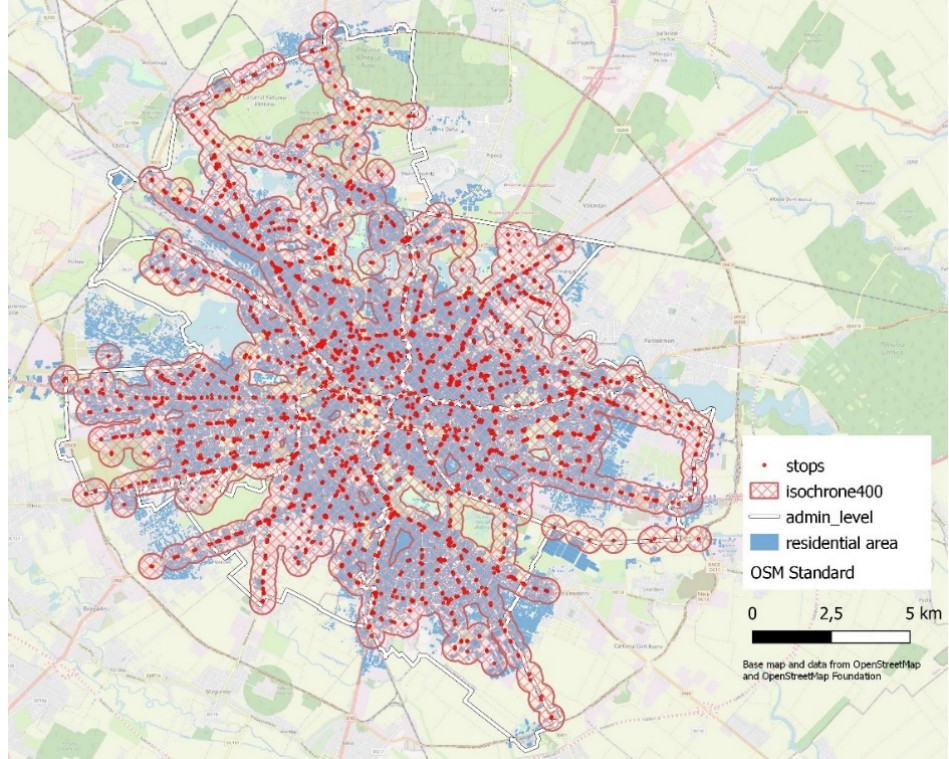

**Figure 8.** Accessibility of public transport stations (isochrone 400 m) (© OpenStreetMap contributors).

Based on the map above, we concluded that, in general, public transport stations can be accessed relatively easily by the populations living in the residential areas, but there are neighborhoods, especially those with low density (single-family homes in the

neighborhoods of the South-Western and North-Western areas of the city or in newly developed districts on the outskirts of Bucharest), where the distance to the nearest station is more than a 400 m walking distance.

To increase the accessibility of the main public transport network and the sustainability of last-mile transport, we recommend a future study for the introduction of additional services. The latest research studies [59–61] reveal that new forms of shared e-mobility services are attractive for people in EU cities with similar socio-demographic characteristics, and are well received in remote areas with limited possibilities for reaching the main public transport network. Therefore, e-micro-mobility services, on-demand e-transport services, or services based on the use of autonomous and connected vehicles can be used. Thus, the use of sustainable means of transport could be encouraged to the detriment of transport by personal cars.

### 3.2. Optimizing the Headway of Vehicles Using the QGIS Software and the PostGIS/PostgreSQL Database

Optimizing the headway of vehicles is based on public transport information (stops and schedule), using a standardized GTFS format, and from the direct observation of the phenomenon that often occurs when two or more trams simultaneously reach the same station: the agglomeration of platforms with unwanted consequences for the transfer time, the loss of the desired line, and safety issues because of crowding, leading to a considerable decrease in the attractiveness of public transport.

The integration and correlation of data were performed for all the tramlines by using open-source software (QGIS and PostGIS), resulting a database with more than 125,000 records, consisting of all the tramline schedules and stations (Figure 9).

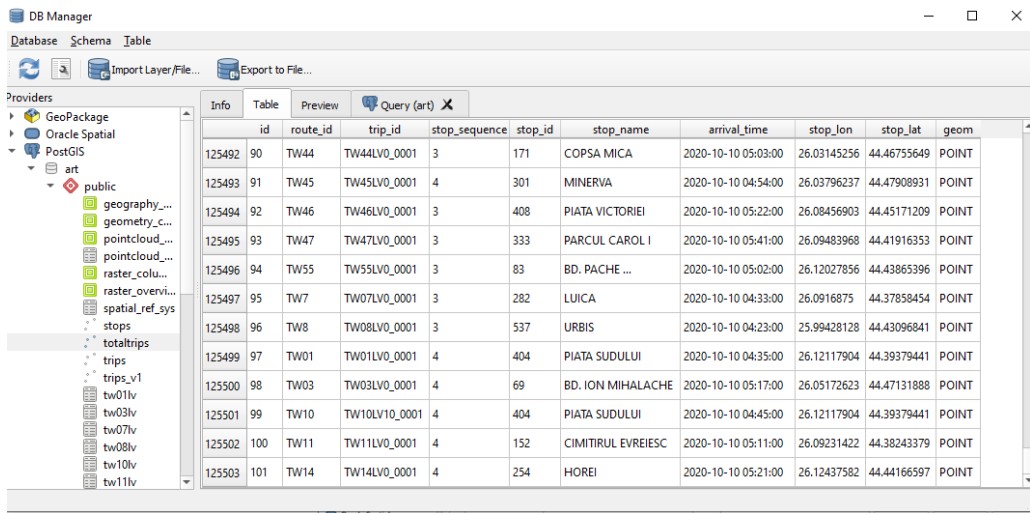

**Figure 9.** Extract from database with all tramlines in Bucharest.

By running the model, the practical observations were confirmed: several vehicles arrived at the same stations at the same times.

The SQL query and its result are presented in Figure 10, where it can be observed that, in 5400 cases, two vehicles arrived at the same station at the same time.

The same algorithm was applied to see if three or four vehicles could reach the same station at the same time, resulting in 348 and 14 cases, respectively (Figure 11). Applying the algorithm for five vehicles returned one case (Figure 12). Having more than three or four tram vehicles at the same station at the same time might affect the operation, when considering the access and egress of vehicles. Additionally, it could affect the image as well as the safety of the passengers on the platform because of disorganized movement, especially when passengers want to switch line.

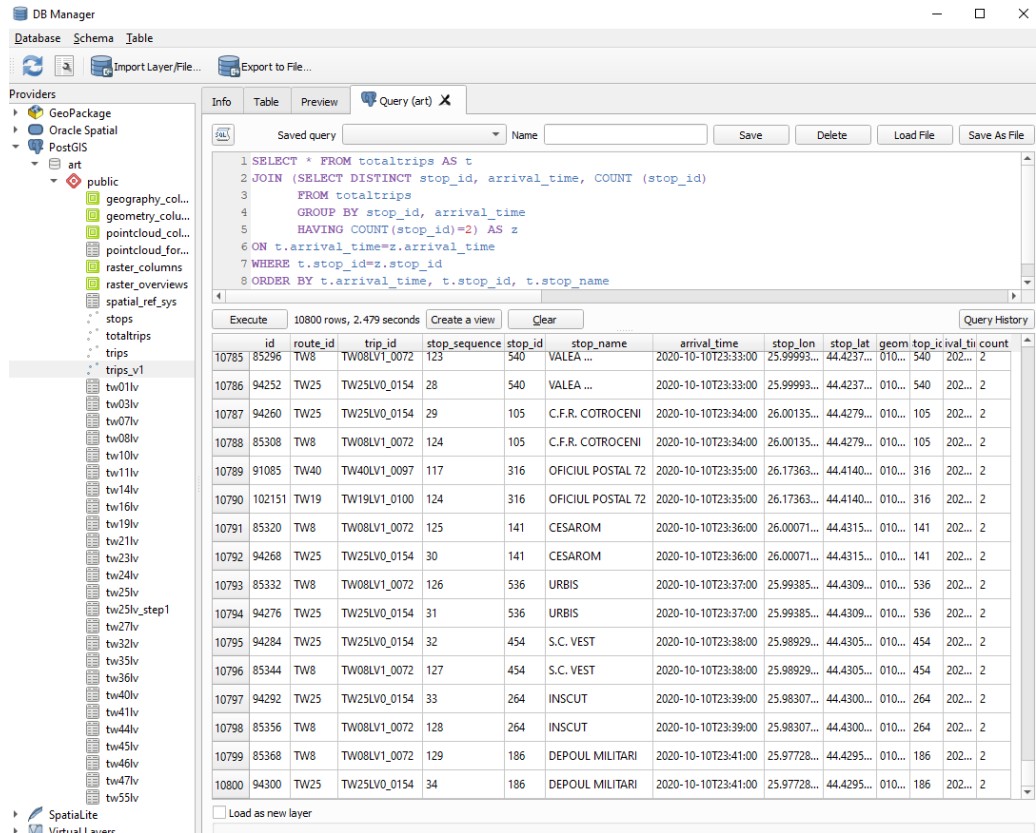

**Figure 10.** Number of cases of 2 trams arriving at the same station at the same time.

By modifying the arrival time from the schedule database and by repeated iterations, headway optimization was achieved. For instance, we chose to modify, by 1 min delay, the tramline number 25 schedule, which appeared as a third vehicle at stop_id 122, together with lines 8 and 11 (Figure 13). It can be observed that, when running the algorithm with the new time values, lines 8 and 11, at the same stop_id and arrival_time, appear in cases where two vehicles arrive at the same station at the same time (Figure 14). The iteration was repeated again after altering the schedules of the remaining tramlines.

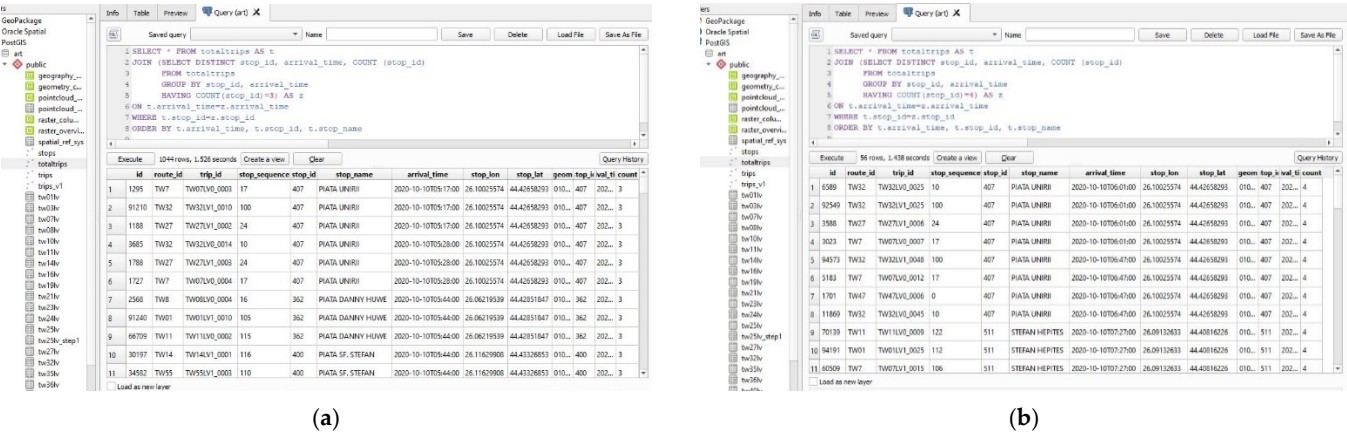

(**a**)                                            (**b**)

**Figure 11.** Number of cases of 3 trams (**a**) and 4 trams (**b**) reaching the same station at the same time.

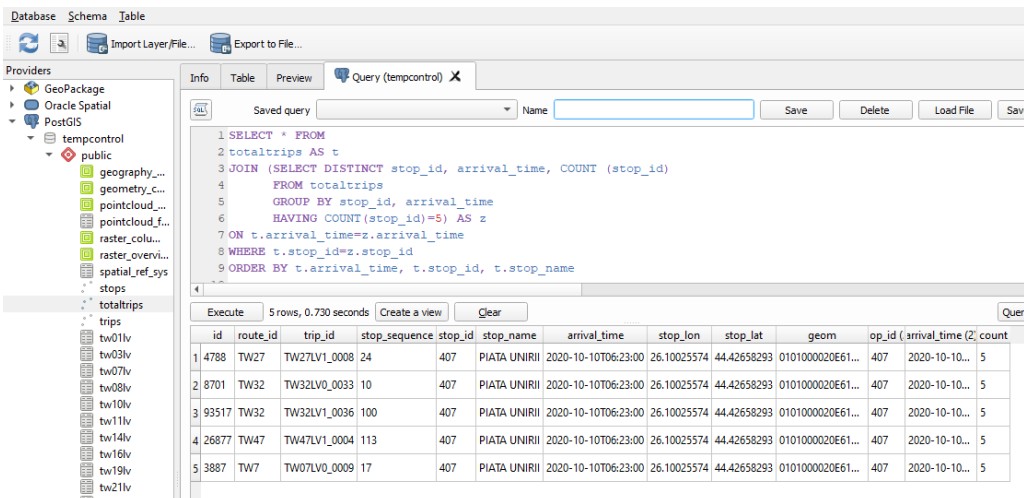

**Figure 12.** Number of cases of 5 trams reaching the same station at the same time.

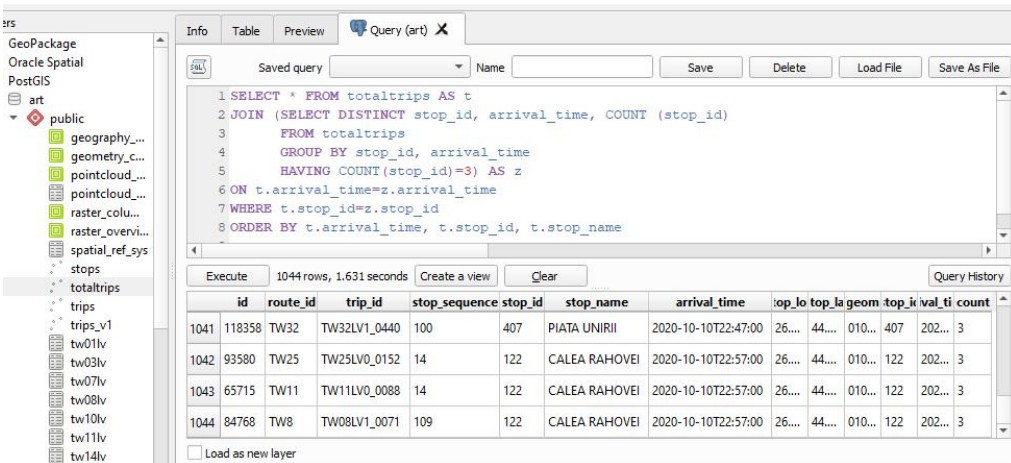

**Figure 13.** Example of 3 vehicles arriving at the stop_id 122 at the same time.

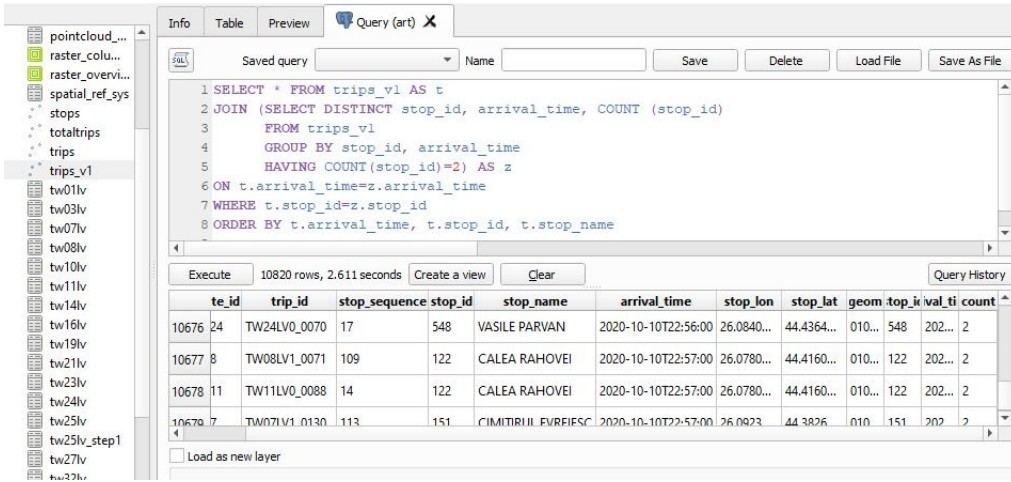

**Figure 14.** No. of vehicles arriving at stop_id 122 at the same time after changes of schedule.

The proposed model allows the visualization of the vehicles' positions on the network by making use of the QGIS Temporal controller feature. By running this feature, we can obtain an animation with the positions of vehicles on the digital map and observe the

situation of tram operation during a specific time range or at a specific time. Several timeframes are depicted in Figure 15.

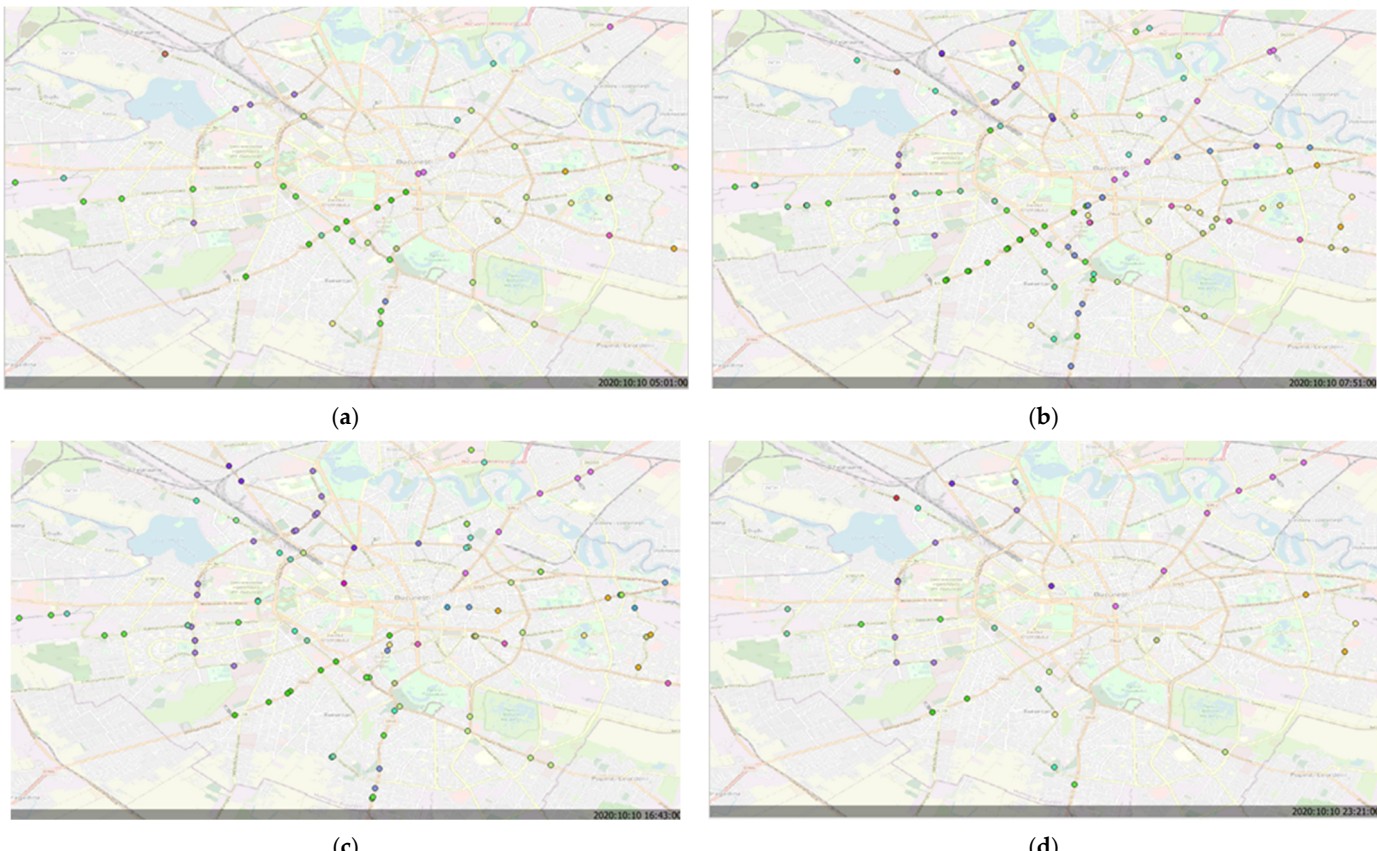

(**a**)

(**b**)

(**c**)

(**d**)

**Figure 15.** Tram position in several timeslots, each color representing the vehicle position for a specific tramline (e.g., light-green points represent the vehicles operating on tram line no. 8; red points, vehicles on tramline 21) (© OpenStreetMap contributors). (**a**) 05:01. (**b**) 07:51. (**c**) 16:43. (**d**) 23:21.

The capacity of the station and the tram schedule can be appropriately designed if the number of passengers accessing and egressing the vehicle is known.

The model can be improved by adding the number of passengers that access/egress from vehicles or the population within the 400 m isochrone. Thus, the model allows the simulation of the vehicle movement in the existing urban public transportation tram network and can support decisions related to optimizing the tram schedule during peak hours. Furthermore, it allows an appropriate design of platforms or the allocation of different-capacity vehicles to adapt the fleet to demand.

### 3.3. Limitations of the Method and Further Research

It should be noted that, during our research, we encountered several problems related to the lack of open data regarding:

1. Standardized public transport data, such as schedules, lines, the GPS positions of the stations, and the number of passengers per means of transportation. For instance, the schedule is available only in PDF format, making the transfer of data in a usable format rather difficult.

2. Population data, consisting of the number of inhabitants and their distribution in the city, the main employers, and the number of workplaces. The most recent available population data were provided by the National Population Census in 2011, and the population number is only available at the district level but not at the level of neighborhoods/residential areas.

The lack of data currently impedes the precise calculation of accessibility for the population living at walking distances from public transport stops/stations.

Romanian cities are not registered on the platform dedicated to Open Mobility Database ("OpenMobilityData—Public Transit Feeds from around the World" n.d.), even though 1247 public transport operators and authorities from 673 cities around the world have included their mobility data on this website. The lack of these data raises difficulties in analyzing the reliability of the public transport network.

The Traffic Analysis Zones used in the Sustainable Urban Mobility Plan 2016–2030—Bucharest-Ilfov Region are based on the data available from the 2011 Census. However, they provide useful information on the modal split in the region: public transport is used for home-based work trips by 33.2% of the population and by 54.9% for home-based education trips.

Under these circumstances, it becomes difficult to correctly estimate the catchment area of a stop/station (the existing population within a walking distance of 400 m) and the passenger flow per station in a specific hourly range.

The study did not consider the introduction of new tramlines, but when they are decided upon, the databases (GIS and schedule) can be easily updated and the iterations can be run under the new circumstances.

The capacities of the vehicles are available, but the numbers of passengers boarding and alighting are not publicly available. For this reason, it is not currently possible to correlate the vehicle capacity with the demand and schedule. The presented model was conceived to allow the addition of other features such as the number of passengers per station and the vehicle capacity.

The developed method could be very convenient for a low number of lines that use a common route segment.

## 4. Conclusions

Public transport and non-motorized travel are essential components of sustainable cities. The information and analyses we present in this paper constitute the basis for research related to urban public transport performance with the support of open tools.

The Bucharest public transport map we developed in QGIS for this research constitutes a standalone tool that can be used to carry out different transport-related studies. In addition, it demonstrates the capabilities of data integration and manipulation with other open-source software (PostGIS and PostgreSQL), allowing numerous transportation and land-use planning analyses.

The method developed for the analysis of public transport performance using open tools has multiple advantages in terms of accessibility, flexibility, cost-efficiency, and the possibility of running it on different operating platforms.

It is noteworthy that the method developed in this paper represents a tool for the analysis of performance for urban public transport systems based on two factors: the accessibility of public transport for residential areas and the optimization of the headway of tram vehicles. More specifically, it will contribute to the easy assessment of the accessibility of public transport for residents in neighborhoods and allow simulations for the correlation of transport schedules and vehicle arrival times in stations to be carried out. In addition, by adding the numbers of passengers that access/egress from a vehicle, the number of vehicles (or their transport capacity) operating in specific time intervals (during peak hours and outside them) can be adapted and optimized, depending on the transport demand.

Based on the study results, decisions can be made by public transport operators to improve and/or reconfigure infrastructure, such as increasing the capacity of transport platforms where the presence of several vehicles/lines is required in the same time slot, in the same station, or in terminals.

Future research into developing this model should attempt to incorporate new elements for the determination of public transport performance (frequency, waiting time, etc.), together with the macro-analysis of the areas that are not well covered by public

transport, for the introduction of micro-mobility services and autonomous vehicles as a feeder for the public transport network. For instance, in the newly built districts or in university campuses, a sustainable solution based on transport on demand using low-capacity autonomous vehicles for last-mile transport, correlated with main the public transport vehicles, could be researched.

Similarly, research can be extended in the case of the introduction of a new tram line for embedding the characteristics of the infrastructure (the width of the street, slopes, green lights, the capacity of the stations, etc.) in the model.

In the medium term, the replication and adaptation of the methodology for other large and medium urban centers in Romania will be considered.

**Author Contributions:** Conceptualization, L.A. and O.L.; methodology, L.A. and O.L; software, L.A.; validation, O.L. and L.A.; formal analysis, L.A. and O.L.; investigation, L.A.; resources, L.A.; data curation, L.A. and O.L; writing—original draft preparation, L.A.; writing—review and editing, O.L. and L.A; visualization, L.A.; supervision, O.L.; project administration, O.L.; funding acquisition, O.L. All authors have read and agreed to the published version of the manuscript.

**Funding:** This research was funded by MINISTRY OF EDUCATION AND RESEARCH through Institutional Development Fund, grant number CNFIS-FDI-2021-0144-INSTRUCT-4.

**Institutional Review Board Statement:** Not applicable.

**Informed Consent Statement:** Not applicable.

**Data Availability Statement:** Not applicable.

**Acknowledgments:** We would like to acknowledge the invaluable support provided by the FDI-INSTRUCT advisory team from UTCB in writing this article.

**Conflicts of Interest:** The authors declare no conflict of interest.

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
