# Peer review of "Open Tools for Analysis of Elements Related to Public Transport Performance. Case Study: Tram Network in Bucharest"

_applsci, doi:10.3390/app112110346_

Round 1

Reviewer 1 Report

I rate the reviewed article very highly.

The authors combined theoretical considerations with practical analysis of public transport system on the example of the city of Bucharest.

The literature review is done in a proper way. Figures are clear and well described in the text.

Therefore, I believe that the article should be published immediately.

Author Response

Thank you very much!

Reviewer 2 Report

The paper describes a methodology for developing a digital map of a city, useful for analyzing some performance indicators of an urban public transport system, and based on the use of open-source software such as QGIS, a free GIS software, and PostgreSQL, an object-relational database system. The authors choose the city of Bucharest as a case study. The paper in the current form is not ready for publication in this journal and must be improved considering the following comments.

  1. Although the methodologies described by the authors are of absolute interest and useful to public transport companies and policymakers, they are well known. Therefore, authors should better explain the novelty of their approach and emphasize which gap in knowledge the paper wants to fill.
  2. The English language needs to be improved.
  3. The abstract is too long, and the authors repeated some concepts several times. It should be summarized, remembering not to exceed 200 words.
  4. In the Introduction, the authors should introduce the research problem, placing the study in the context of the existing literature to emphasize its novelty, originality, and advancement in the research. Therefore, in my opinion, the literature review should be included in the Introduction, in order to emphasize the novelty of the approach.
  5. The research question is clear and general, applicable in all urban contexts. However, it is presented after having already introduced the Bucharest case study. As the developed methodology can be useful for public transport companies and policymakers in different cities, the authors should introduce the Bucharest case study only after outlining the research question, highlighting how the public transport system of Bucharest, Romania, was chosen as a case study to answer the research question.
  6. In the Introduction, the authors state that "Urban areas with a good public transport supply, also favor pedestrian and bicycle travel" but this is partially true. This statement is only true if a safe and comfortable infrastructure for cyclists and pedestrians exists. For example, poor walkability of the pedestrian environment can affect the propensity to use public transport as highlighted in: D'Orso, G., Migliore, M., 2020. A GIS-based method for evaluating the walkability of a pedestrian environment and prioritized investments ". Journal of Transport Geography, 82, 102555.
  7. Please refer to QGIS as a "software" rather than a "program".
  8. Please include an outline of the paper in the Introduction.
  9. “Materials and Methods” and “Results and discussion” are quite confusing. I suggest rewriting them considering the following comments. In "Materials and Methods" please briefly describe the software used in the methodology, specifying their characteristics, the plugins needed, the version used. The goal of the methodology is to develop a digital map of Bucharest that allows you to carry out analyses on the performance of public transport. Therefore, in “Materials and Methods”, the authors should also describe the steps of the development of the digital map and all the analyses that can be carried out with it. In "Results and discussion", instead, you can underline the results of these analyses (results in terms of capillarity, accessibility, timetables), the opportunities and limitations of the methodology, critically comparing it with the other existing methods, evaluating its strengths and weaknesses in relation to them.
  10. I didn’t understand how the review of the literature has been of support in the definition of the methodology presented in “Materials and Methods”. Please explain this step better or, as I already suggested, move the literature review to the Introduction.
  11. The literature review in this form is simply a list. The literature review is necessary for you to clarify the “contribution” of your study and to place it in the right context. In general, the authors should present the specific debate for their study. In the current form, the authors failed to present the study debates and failed to discuss the debates. For instance, you can write: “X et al. (2011) showed that……”. “Y (2005) also showed that ………”. “The controversy is ...” “The limitations of the methodology proposed by X are…”
  12. In line 149, please correct “PosrgreSQL” (“PostgreSQL”).
  13. In line 158, please correct “programs” (“software”) and “Open Street” (“OpenStreetMap”).
  14. In line 167, please correct “an integrated approaches”.
  15. In line 170, “was examined” is repeated twice.
  16. The methodologies used are not explained in detail and, therefore, cannot be replicated by other researchers. Please describe all the steps of the methodology, and for each of them the software used, the operations performed (if you think it is appropriate, even the scripts in SQL language), the plugins used.
  17. How was OpenStreetMap used in the case study? Has only the road network been extracted? Or also the buildings, the railways, and the administrative units? What are they used for?
  18. In line 210, the authors said that “Localization of the stations on the map was performed manually”. What do you mean by "manually"? Did you use a background map such as OpenStreetMap or Google Earth on QGIS and locate stations finding their position on the map? In my opinion, this is the simplest method to have a fairly accurate location of the stations, so the coordinates provided by the transport company are not necessary.
  19. PostGIS is a PostgreSQL extension that allows the database to connect with GIS software. Please clarify that PostGIS and PostgreSQL are not the same.
  20. Please, specify the main aspects of the public transport systems. How many tram stations/ bus stops are there? How many lines? How many vehicles? How many passengers per day?
  21. Are the public transport systems in the city (tram, bus, trolley) integrated? Is there a fare integration? A physical integration? Integration of the timetables?
  22. In line 260, you specify that "Based on the present map, fast decision on reconnection may be taken by the public transport operator." This observation is partially true because infrastructural constraints must also be considered. For example, for the introduction of a new tram line, the width of the roads, the slope and other constraints must be considered. These additional attributes could still be displayed on the GIS software to find those roads that can accommodate the tram system.
  23. What operation in QGIS was performed to determine the isochrones? Why did you use radial isochrones rather than the more precise road network-based isochrones?
  24. The authors didn’t give a precise measure of the accessibility of the public transport system. Looking at the map, the readers can estimate the capillarity of the system in the territory, but they can’t know if the neighbourhoods outside the catchment area are the most populous in the city or not, not being able to understand if the accessibility of the public transport system is good or poor. By combining demographic data with the accessibility map, it is possible to determine the share of the population living at a walking distance from public transport stops (and the socio-demographic characteristics), as well as that which does not fall within the public transport catchment area. This would give the reader greater awareness of the effective accessibility of the public transport system. You can use data from the National Population Census (2011) specifying that they are not up to date.
  25. In line 281, the authors said that “The latest research studies [50], [51], [52] revealed that new forms of shared e-mobility service are attractive for the people with comparable socio-demographic characteristics”. Comparable to what?
  26. Please, better describe the steps that led to the optimization of the headway of trams and how this analysis is linked to the digital map of Bucharest. It seems that it is more related to some queries made on a PostgreSQL database created from GTFS files rather than the digital map.
  27. In line 301, the authors said that two vehicles arrive at the same station at the same time in 10600 cases. In Figure 8, however, the table has 10800 rows. However, the times that two vehicles arrive at the same station at the same time are not 10800 but 5400 (= 10800/2) since each row represents a vehicle arriving at a certain tram stop at a certain time.
  28. Similarly, the cases in which 3, 4, and 5 vehicles arrive at a station at the same time are respectively 348 (= 1044/3), 14 (= 56/4), and 1 (= 5/5).
  29. In figures 8 and 9 please make “id”, “stop_id” and “arrival_time” columns more visible since from these three fields the reader can understand how different lines (id) arrive at the same station (stop_id) at the same time (arrival_time).
  30. Describe Figure 11 better. What do the points in different colours represent?
  31. In line 348, the authors said that “it becomes difficult to correctly estimate the right position of the stations” but the right position of the stations can be estimated simply by finding stops and stations on a background map like Google Earth on QGIS, or through site inspections or Google Street View.

Author Response

Dear Reviewer, 

Thank you very much for your time, effort and interest in our research. Please see the attachment.

Reviewer 3 Report

Dear Authors,

Thank you for submitting a very interesting article to the MDPI Journal. I believe that you have made a lot of effort on a very interesting topic.

The subject of the article fits into a properly defined research problem, to which properly defined goals were subordinated. The article has a clear structure, individual fragments of the article merge into a logical whole. The article is a significant contribution to the development of science, but it can also be an interesting source for decision-making centers in the operation of tram networks.

The article has minimal shortcomings that do not contribute to a very positive assessment.

Deficiencies that need to be corrected (to be verified by the editorial team as they are more editorial than factual and methodological flaws).

  1. The titles of figures 7 and 8 should be transferred to the figures (they are currently on a separate page)
  2. Improve the citation style in the manager, it is different from the standard in MDPI.
  3. The year is missing in some items - please specify (can be specified).
  4. At the end of journal titles, a comma appears before closing quotation marks.
  5. For the year of publication, the month is rather not given.
  6. All the authors are missing from the references in some items (eg 10, 29) - please verify and name all the authors.
  7. In some items the order is reversed, there should be a surname first, then a first name (other entries for verification).
  8. In the references [29] there is a full name, there should be an initial - please check the others that the surname and the first name have not been mixed up.
  9. Web pages should be given in a slightly different way: first the name / title of the page, then the link and date of access. A link to the page is not enough.

Good luck!

Greetings

Reviewer

Author Response

Dear reviewer,

Thank you very much for your interest for our research. Please see the attachment.

Round 2

Reviewer 2 Report

The authors have followed my previous suggestions. Thus, the manuscript has been sufficiently improved to warrant publication in Applied Sciences. However, to ensure that the paper is ready for publication, the authors must make the following minor revisions:

  1. please make the entire contents of the "arrival_time" column readable in Figs. 10, 11a, 11b and 13, as you have already done for Figs. 9 and 12.
  2. please improve the image quality of Fig. 14.
  3. please specify how did you modify the tramline number 25 schedule. From the arrival time 22:...... to the arrival time 23:15:00? Is this change in schedule feasible for the transport company?

Author Response

Thank your your comments and suggestions. Please find enclosed the attachment. 
